# The validity of mid-upper arm circumference as an indicator of underweight, overweight and obesity adults in Bangladesh

Sheikh Arafat Rahman🔗☺, Md. Humayan Kabir🔗☺, Shaikh Shahinur Rahman, Md Kamruzzaman🔗*

Department of Applied Nutrition and Food Technology, Islamic University, Kushtia, Bangladesh

☺ These authors contributed equally to this work.
* mkzaman.m@gmail.com, md.kamruzzaman@anft.iu.ac.bd

## Abstract

### Background

Mid-upper-arm circumference (MUAC), as a simple measurement, is an effective alternative to body mass index (BMI) for resource-limited countries such as Bangladesh. The present study aimed to investigate the age- and sex-specific MUAC cut-off points as screening tools for underweight, overweight and obesity in Bangladesh.

### Subject and methods

A nationally representative dataset (BDHS-2017/18) comprising of 12,962 adults individuals (females: 56.5%, mean age: 39.5±16.0 Years, BMI: 22.4±4.1 kg/m² and MUAC: 27.4±3.1 cm) was analyzed. BMI was categorized as underweight (BMI <18.5 kg/m²), overweight (BMI: 23 kg/m²), or obese (BMI: ≥27.5 kg/m²). A receiver operating characteristic curve analysis was conducted to determine the optimal MUAC cut-off, based on Youden Index, for underweight, overweight, and obese individuals. Multiple linear regression analysis was performed, adjusting for age and sex, to explore age and sex-specific changes in the association between MUAC and BMI. Restricted cubic spline and binary logistic regression analyses were performed to evaluate the risk of underweight, overweight and obesity among different MUAC quartiles.

### Results

For males, MUAC cut-offs were ≤26.5 cm for underweight (AUC: 82.1%, sensitivity: 71.1%, specificity: 77.8%), ≥28.5 cm for overweight (AUC: 84.8%, sensitivity: 77.6%, specificity: 76.9%), and ≥29.5 cm for obesity (AUC: 89.4%, sensitivity: 87.8%, specificity: 76.6%) in younger and middle-aged individuals; older males had cut-offs of ≤25.5 cm, ≥27.5 cm, and ≥29.5 cm respectively. For females, the cut-offs

**Data availability statement:** Third party data was obtained for this study from the DHS Program. Data may be requested from the DHS Program after creating an account and submitting a concept note. More access information can be found on the DHS Program website (https://dhsprogram.com/data/Access-Instructions.cfm). The authors confirm that they had no special access privileges that others would not have.

**Funding:** The author(s) received no specific funding for this work.

**Competing interests:** The authors have declared that no competing interests exist.

were lower: ≤25.5 cm for underweight in younger and middle-aged, and ≤24.5 cm in older; ≥27.5 cm for overweight in younger, and ≥26.5 cm in middle-aged and older; and ≥28.5 cm for obesity in younger and middle-aged, and ≥27.5 cm in older. The MUAC-BMI correlation was strong (R = 0.69, P < 0.001), stronger in males (R = 0.75) than females (R = 0.69). The MUAC cut-off for underweight, overweight, and obesity slightly decreased with age in both sexes.

## Conclusion

Age- and sex-specific MUAC cut-offs are effective for screening underweight, overweight and obesity in Bangladesh.

---

## Introduction

Anthropometric measurements are non-invasive and well-established methods for assessing nutritional status and body weight changes in individuals and populations [1]. The WHO recommends anthropometry as the most portable, universally acceptable, and inexpensive procedure for measuring body composition and health status [2]. Owing to its cost-effectiveness and portability, anthropometry is particularly useful for individuals from low-to middle-income countries (LMICs); where advanced assessment techniques may not be appropriate because of their affordability.

Body mass index (BMI), measured using weight and height, is an appropriate indicator of the nutritional status of adults. BMI is consistently correlated with underweight or undernutrition as well as overweight and obesity [3]. Although BMI appears to be simple and ideal for population level studies, it often fails to discriminate between leanness and fatness. The risk largely depends on sex and ethnicity, which are disregarded in the measurement and classification of BMI [4]. Moreover, accurate weight and height measurements are necessary, and require proper instrument maintenance and calibration. Field studies also face challenges including logistic constraints such as limited access to remote areas and transportation issues and sampling bias, which can affect the reliability and representativeness of the results.

Mid-upper arm circumference (MUAC) is a simple, quick and effective anthropometric procedure. It is measured at the midpoint of the upper arm between the shoulder (acromion) and elbow (olecranon) [5]. Unlike BMI, MUAC requires no calculations, and utilizes only a standardized tape. It is sensitive and specific for reflecting arm muscle area and undernutrition in children [6]. However, MUAC many not be accurate in individuals with edema or muscle hypertrophy and may exhibit limited sensitivity to incremental changes in body composition. Decreased MUAC is strongly associated with mortality in adults [7] and underweight or energy deficiency in children [6]. Changes in MUAC often reflect changes in muscle mass and subcutaneous fat with high precision [5]. Numerous studies have shown that MUAC is a convenient tool for children [6], women, and ambulatory hospitalized patients [8]; where weight and height are not measurable.

Previous studies have reported a strong association between MUAC and BMI; though, ethnic diversity in overall and regional body fat is significant. For instance, the global cut-off point for undernutrition is < 24 cm [9], while national thresholds vary: < 25.4 cm in South Sudan [10], < 24.5 cm in Nepal [11], < 23 cm in South Africa [12], sex specific values of <23 cm (males) and <22 cm (females) in India [13]. For overweight and obesity, recommended MUAC cut-offs include ≥26.91 cm in Ethiopia [14], ≥ 28.5 cm in South India [15] and ≥27.95 cm in South Africa [16]. These variations emphasize the need to establish ethnic- and region-specific cut-off points.

Bangladesh, an LMIC, has undergone rapid socioeconomic, demographic, and nutritional transition. The country currently faces a dual burden of malnutrition, with 17.1% of the population being underweight and 40.2% overweight and obese [17]. Overweight and obesity has increased sharply due to urbanization and the spread of processed foods, leading to increased risk of metabolic disorders such as diabetes and hypertension. Thus, early screening using cost-effective anthropometric tools is necessary.

In Bangladesh, MUAC has received limited attention as an alternative to BMI for identifying individuals with underweight, overweight, and obesity. One study attempted to propose MUAC cut-off for underweight screening, but was unable to generalize its findings due to the limited geographical coverage. This limitation underscores the need for further validation studies [18]. Consequently, the present study was conducted to explore age- and sex-specific MUAC cut-offs in adults for screening underweight, overweight, and obese individuals in a large, representative sample in Bangladesh.

## Methods

### Study design, participants and settings

This study used secondary data from the Bangladesh Demographic and Health Survey (BDHS) 2017/18, collected from October 2017 to March 2018. The BDHS is a nationally representative survey conducted by the National Institute of Population Research and Training (NIPORT), with technical assistance from ICF International and financial support from the United States Agency for International Development [19]. For the BDHS survey, the study sample was selected, using a two-stage sampling procedure. Primary sampling units (PSUs) were selected in the first stage based on a probability proportional to size (PPS) approach. In the second stage, households were selected from each PSU using a systematic sampling. The survey encompassed 20,250 households across 675 clusters covering both rural and urban areas. Clusters were chosen in the first stage, and 30 households per enumeration unit were selected in the second stage. Initially the BDHS 2017 dataset included 55,273 adults aged ≥18 years. However, 42,311 individuals with missing MUAC and BMI were excluded from the dataset. Consequently, data pertaining to 12,962 adult individuals (aged ≥18 years; 5639 males and 7323 females (nonpregnant)) were extracted and utilized in the present study.

### Ethics approval

The data were secondary; therefore, formal ethical approval was not required. A detailed ethical procedure is available in the BDHS report [20]. The study was approved by the National Research Ethics Committee (NREC) of the Bangladesh Medical Research Council (BMRC) (Ref: BMRC/NREC/2016–2019/324). This study was conducted in accordance with the principles of The Declaration of Helsinki (2008), and all participants provided written informed consent. Electronic approval for the use of the dataset for this study was received from ICF International in January 2024.

### BMI categories

In this study, BMI was used as the response variable. The raw BMI data from the dataset were categorized according to the WHO-recommended BMI category for Asian people to classify the participants as follows: BMI < 18.5 kg/m$^2$ as underweight or energy deficient, 18.5–22.99 kg/m$^2$ as normal weight, 23.0–27.49 kg/m$^2$ as overweight and ≥27.5 kg/m$^2$ as obese [21].

## Explanatory variables

MUAC was utilized as the primary explanatory variable. MUAC was measured at the midpoint of the arm, between the olecranon and acromion, following standard procedures. Weight was measured using lightweight electronic scales with a digital display, ensuring minimum clothing. In contrast, height was measured in the standing position using a stadiometer [22]. Two replicate measurements of MUAC, weight and height were obtained, and the mean of the two measurements was recorded. Each study team comprised two members, one male and one female, and the study team received appropriate training to measure MUAC, weight and height. The anthropometric measurement and questionnaire were pretested in 100 households, and corrections were implemented based on observations from the pretest field visit and suggestions made by the study team [19]. Additional variables, such as age, sex, and area of residences were included in this study. Age was categorized as younger (18–40 years), middle (40–60 years), or older (>60 years). The age group (18–40, 40–6- and >60 years) were selected on a combination of epidemiological and alignment with previous studies that investigated age-related difference in body composition and nutritional risk [23,24]. These categories represent significant physiological transitions in early adulthood (18–40), middle (40–60) and older adulthood (>60), which are frequently associated with alterations in muscle mass, fat distributions, and metabolic risk. Individuals were classified as having diabetes if they exhibited a fasting plasma glucose level ≥7.0 mmol/L and/or were receiving treatment with diabetes medication [25]. Similarly, a person was considered hypertensive if they had systolic blood pressure levels ≥140 mmHg and/or diastolic blood pressure levels ≥90 mmHg and/or were undergoing treatment with hypertensive medication [26,27].

## Statistical analysis

To summarize the descriptive statistics, categorical variables were presented as frequencies and percentages, while continuous variables were presented as mean±standard deviation. Visual methods using a Q-Q plot and Shapiro-Wilk's test were performed to assess the normality of the data, and the data met the criteria for normality. An independent samples t-test was performed to identify significant differences between males and females. Pearson's correlation coefficient was calculated to examine the linear relationship between MUAC and BMI. Multiple linear regression was performed at a higher order of BMI and age to determine the relationship between BMI and MUAC at different ages and BMI levels. To determine the optimal MUAC cut-off point for identifying underweight, overweight, and obese individuals, receiver operating characteristic (ROC) curve analysis was performed in R. This analysis was conducted for the entire population and as well as for all subgroups stratified by sex and age. The area under the curve (AUC) was utilized to measure the overall accuracy of the MUAC cut-off points for identifying underweight, overweight, and obese individuals [28]. ROC curves were generated in RStudio using the R packages "pROC" and "AUC". The ROC curve is a graphical representation in which sensitivity is displayed on the Y-axis and 1-specificity is displayed on the X-axis. The area under the curve (AUC) was subsequently calculated, and the Youden index (YI) was employed to obtain the optimal cut-off points for MUAC. The diagnostic ability and predictive value of the MUAC cut-off points against BMI were evaluated using sensitivity, specificity, positive predictive value, negative predictive value, and 95% confidence intervals. A restricted cubic spline (RCS), with 7 knots, was used to measure the odds of being underweight, overweight and obese according to their MUAC. This provided a better model fit with lowest Akaike Information Criterion (AIC). The RCS figure was generated in RStudio using R packages "plotRCS". RCS is efficient in addressing the nonlinear relationship between continuous and response variables, while also possessing the capability to locate crucial key points. MUAC was divided into quartiles and binary logistic regression analyses were conducted to evaluate the likelihood of being underweight, overweight, or obese among individuals in different MUAC quartiles. MUAC quartile 4 was considered a reference for underweight, and quartile 1 for overweight and obesity. The logistic regression model was adjusted for age, sex, and area of residence. Using descriptive statistics, the prevalence of general morbidity such as diabetes and hypertension was calculated based on our proposed MUAC cut-off. The significance level was set at p<=0.05. All analyses were conducted using RStudio software (R 4.4.1).

## Results

### MUAC and BMI by age and sex

Table 1 presents the demographic and health-related variables of the study population. The overall mean±SD age of the participants was 39.5±16.0 years, and males were older than females were (41.4±16.4 vs 37.9±15.6 years, P<0.001). More than half of the participants were aged 18–40 years (60.2%), and more than half of the participants were females (56.5%). The mean weight of the total study participants was 54.3±11.3 kg, with males having a greater mean weight than females. The mean height of the study population was 155.6±8.78 cm. On average, males were taller than females (P<0.001). The mean BMI of the total study population was 22.4±4.1 kg/m² and compared with males, females had a greater mean BMI (P<0.001). The MUAC was greater in males than in females (P<0.001). According to the body mass index (BMI), the proportion of individuals with a normal BMI was the highest (42.6%), followed by overweight (29.4%), underweight (17.0%), and obese (11.0%) individuals (Table 1).

**Association between MUAC and BMI.** The MUAC was positively associated with BMI (R=0.69, P<0.001), and the association was stronger in females (R=0.75, P<0.001) than in males (R=0.69, P<0.001) (Fig 1A). Multiple linear regression demonstrated that MUAC is slightly lower at higher orders of age and higher orders of BMI (S1 Table).

### MUAC cut-offs by age and gender

The efficacy of MUAC in accurately identifying underweight, overweight, and obesity was evaluated using the ROC curve (Fig 1B). Sensitivity (SENS), specificity (SPEC), positive predictive value (PPV), and negative predictive value (NPV) were derived from the ROC curves analyses of MUAC. The overall optimal MUAC cut-off points for underweight, overweight, and obese are presented in Table 2. The optimal cut-off points for detecting underweight was ≤25.5 cm in the entire sample, irrespective of age and sex, with an AUC of 84.0% (95% CI: 83.4–84.6). The optimal cut-off points for men and women were ≤26.5 cm (AUC: 83.1%) and ≤25.5 cm (AUC: 87.9%), respectively. The optimal MUAC cut-off points for younger individuals (≤26.5 cm) were greater than those for older and middle-aged individuals

**Table 1. Overall and gender-specific characteristics of the study participants.**

| | All (n=12962) [Mean±SD or n(%)] | Male (n=5639, 43.5%) [Mean±SD or n(%)] | Female (n=7323, 56.5%)[Mean±SD or n(%)] | P† |
|---|---|---|---|---|
| **Age (Years)** | 39.5±16.0 | 41.4±16.4 | 37.9±15.6 | <0.001 |
| 18-40 Yrs | 7793 (60.2) | 3189 (56.7) | 4604 (63.0) | <0.001 |
| 40-60 Yrs | 3210 (24.8) | 1384 (24.5) | 1826 (25.0) | >0.05 |
| ≥60 Yrs | 1934 (15.0) | 1054 (18.8) | 880 (12.0) | <0.001 |
| **Weight (kg)** | 54.3±11.3 | 57.4±11.0 | 51.8±10.9 | <0.001 |
| **Height (cm)** | 155.6±8.78 | 162.7±6.6 | 150.1±5.9 | <0.001 |
| **BMI (kg/m²)** | 22.4±4.1 | 21.7±3.6 | 23.0±4.4 | <0.001 |
| Underweight (<18.5 kg/m²) | 2201 (17.0) | 1079 (19.2) | 1111 (15.2) | <0.001 |
| Normal weight (18.5–23.0 kg/m²) | 5522 (42.6) | 2694 (47.9) | 2818 (38.5) | <0.001 |
| Overweight (23.0–27.49 kg/m²) | 3815 (29.4) | 1492 (26.5) | 2319 (31.7) | <0.001 |
| Obesity (≥27.5 kg/m²) | 1424 (11.0) | 362 (6.4) | 1062 (14.6) | <0.001 |
| **MUAC (cm)** | 27.4±3.1 | 27.9±2.9 | 27.0±3.3 | <0.001 |
| Quartile 1 (≤25 cm) | 3562 (27.4) | 1118 (19.8) | 2444 (33.4) | <0.001 |
| Quartile 2 (>25–27 cm) | 3306 (25.5) | 1447 (25.7) | 1859 (25.4) | >0.05 |
| Quartile 3 (>27–29 cm) | 3045 (23.5) | 1556 (27.6) | 1489 (20.3) | <0.001 |
| Quartile 4 (>29 cm) | 3049 (23.5) | 1518 (26.9) | 1531 (20.9) | <0.001 |

†: P is from independent sample Student's t-test; Yrs: Years, BMI, body mass index; MUAC: mid-upper arm circumference.

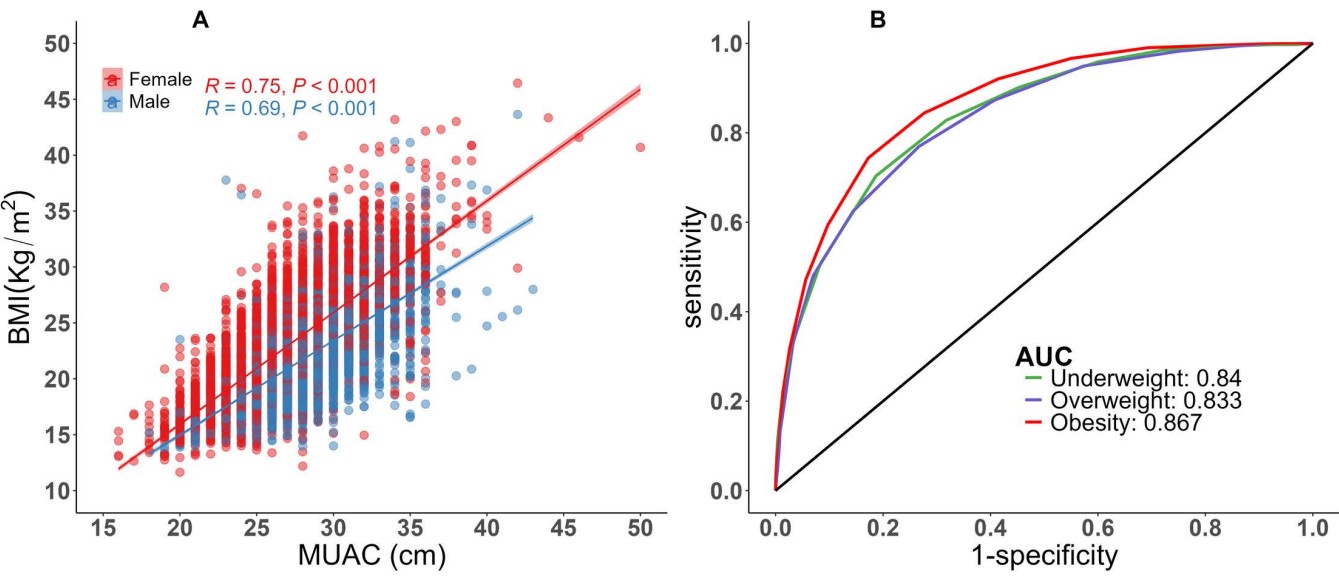

**Fig 1. Correlations between the MUAC and BMI (A.) and receiver operating characteristic curve for mid-upper arm circumference (MUAC) for the detection of individuals with underweight, overweight and obesity (B.).**

(≤25.5 cm). The overall optimal cut-off points of the MUAC for accurately detecting overweight and obese individuals were ≥27.5 cm and ≥ 29.5 cm, with AUCs of 83.3% and 86.7%, respectively. For males, this cut-off points for detecting overweight were greater than that for females (≥28.5 vs ≥ 26.5 cm), while these cut-off points were identical across all age groups (≥27.5 cm). Similarly, the overall MUAC cut-off point for accurately detecting obesity was greater in males than in females (≥29.5 cm vs ≥ 28.5 cm) and in younger individuals (≥29.5 cm) than in middle-aged and older individuals (≥28.5 cm).

### MUAC cut-offs by age in males and females

The MUAC cut-off points were further analyzed based on age category in males and females separately and are presented in Table 3. Across the three age categories, the cut-off points were higher in males compared than in females for screening underweight (younger and middle age: ≤ 26.5 vs ≤ 25.5 cm and older age: ≤ 25.5 vs ≤ 24.5 cm). Similarly, the MUAC cut-off for overweight and obesity was also higher in males than females across all age categories. Furthermore, for both sexes, the MUAC cut-off points were slightly lower for older individuals than for younger and middle-aged individuals (Table 3).

### Prevalence and risk of underweight, overweight and obesity by MUAC cut-offs

The prevalence as well as the risk of being underweight, overweight, and obese by MUAC, performed by RCS and logistic regression analysis (adjusted for age, sex, and area of residence), are shown in Fig 2 and S2 Table. The risk of being underweight increased as MUAC decreased, and the risk of being overweight or obese increased as MUAC increased (Fig 2). RCS proposed similar MUAC cut-offs for underweight (≤25.5 cm), overweight (≥27.5 cm) and obesity (≥29.5 cm).

**Restricted cubic spline model was adjusted for age, gender and area of residences.** Participants with a MUAC within the first quartile (≤25 cm) had a 68.2 times greater risk of being underweight than participants in the fourth quartile. Females within the 1st MUAC quartile had almost twice the risk of being underweight than males in the same quartile

**Table 2. Optimal MUAC cut-off points by gender and age category for screening individuals with underweight, overweight and obesity.**

| | | Underweight (BMI<18.5) | Overweight (BMI≥23) | Obesity (BMI≥27.5) |
|---|---|---|---|---|
| **Total** | *Optimal cut-off point* | *≤25.5 cm* | *≥27.5 cm* | *≥29.5 cm* |
| | AUC (%) (95% CI) | 84.0 (83.4-84.6) | 83.3 (82.7-84.0) | 86.7 (86.1-87.3) |
| | Sensitivity (%) (95% CI) | 70.4 (68.4-72.3) | 77.0 (75.8-78.1) | 74.4 (72.0-76.6) |
| | Specificity (%) (95%CI) | 81.3 (80.5-82.0) | 73.3 (72.3-74.3) | 82.8 (82.1-83.4) |
| | Youden Index | 0.5167 | 0.5026 | 0.5712 |
| | +PV (%) | 6.9 | 17.6 | 3.7 |
| | -PV (%) | 56.5 | 33.8 | 65.3 |
| **Male** | *Optimal cut-off point* | *≤26.5 cm* | *≥28.5 cm* | *≥29.5 cm* |
| | AUC (%) (95% CI) | 83.1 (82.1-84.1) | 84.5 (83.5-85.4) | 88.9 (88.1-89.7) |
| | Sensitivity (%) (95% CI) | 73.2 (70.5-75.9) | 75.5 (73.6-77.5) | 85.4 (81.3-88.8) |
| | Specificity (%) (95%CI) | 77.5 (76.3-78.7) | 77.8 (76.4-79.1) | 77.1 (75.9-78.2) |
| | Youden Index | 0.5069 | 0.5333 | 0.6245 |
| **Female** | *Optimal cut-off point* | *≤25.5 cm* | *≥26.5 cm* | *≥28.5 cm* |
| | AUC (%) (95% CI) | 87.9 (87.1-88.6) | 86.5 (85.7-87.3) | 88.5 (87.7-89.2) |
| | Sensitivity (%) (95% CI) | 84.6 (82.4-86.5) | 83.0 (81.7-84.3) | 80.9 (78.3-83.1) |
| | Specificity (%) (95%CI) | 75.8 (74.8-76.8) | 72.1 (71.0-73.5) | 79.4 (78.5-80.4) |
| | Youden Index | 0.6037 | 0.5511 | 0.6028 |
| **Age 18–40 Years** | *Optimal cut-off point* | *≤26.5 cm* | *≥27.5 cm* | *≥29.5 cm* |
| | AUC (%) (95% CI) | 83.3 (82.5-84.1) | 83.3 (82.4-84.1) | 86.7 (85.9-87.5) |
| | Sensitivity (%) (95% CI) | 81.1 (78.8 - 83.3) | 77.7 (76.2-79.1) | 75.8 (72.8 - 78.7) |
| | Specificity (%) (95%CI) | 69.3 (68.3-70.5) | 72.1 (70.8-73.4) | 81.9 (81.0 - 82.8) |
| | Youden Index | 0.5047 | 0.4982 | 0.5774 |
| **Age 40–60 Years** | *Optimal cut-off point* | *≤25.5 cm* | *≥27.5 cm* | *≥28.5 cm* |
| | AUC (%) (95% CI) | 84.9 (83.6-86.1) | 82.7 (81.3-84.0) | 85.1 (83.8-86.3) |
| | Sensitivity (%) (95% CI) | 70.9 (66.6-74.9) | 76.9 (74.6-79.0) | 83.1 (79.3-86.5) |
| | Specificity (%) (95%CI) | 82.9 (81.4-84.3) | 71.4 (69.2-73.5) | 70.4 (68.7-72.1) |
| | Youden Index | 0.5377 | 0.4826 | 0.5355 |
| **Age ≥60 Years** | *Optimal cut-off point* | *≤25.5 cm* | *≥27.5 cm* | *≥28.5 cm* |
| | AUC (%) (95% CI) | 83.0 (81.2-84.6) | 83.7 (82.0-85.3) | 89.2 (87.7-90.5) |
| | Sensitivity (%) (95% CI) | 77.3 (73.5-80.9) | 73.2 (69.4-76.8) | 82.0 (74.4-88.1) |
| | Specificity (%) (95%CI) | 73.5 (71.1-75.8) | 79.4 (77.2-81.5) | 79.7 (77.7-81.5) |
| | Youden Index | 0.5083 | 0.5267 | 0.6163 |

BMI: body mass index; MUAC: mid-upper arm circumference; AUC: area under the curve; +PV: Positive predictive value; -PV: Negative predictive value.

(OR= 136.5 [64.8–287.9] vs. 65.6 [44.0–97.7]). Similarly, older participants (≥60 Yrs.) within the 1st MUAC quartile had a greater risk of being underweight than middle-aged and younger participants within the same 1st MUAC quartile. The prevalence of overweight and obesity increased as the MUAC increased. Participants with MUAC levels within the fourth quartile (>29 cm) exhibited a 57.9-fold increased likelihood of being overweight compared to participants with MUAC levels within the 1st quartile, with a higher risk among females (OR: 121.43 vs 89.6) and younger participants (OR: 59.0 vs. 43.4). Similarly, participants with MUAC within the fourth quartile (>29 cm) demonstrated a 143.3-fold greater risk of obesity compared to participants within the 1st MUAC quartile, with a higher risk among females (OR: 194.6 vs. 142.6) and younger participants (OR: 143.3 vs. 119.0) (S2 Table).

**Table 3. Optimal MUAC cut-off points by gender in different age categories for screening individuals with underweight, overweight and obesity.**

| | | | *Underweight (BMI < 18.5)* | *Overweight (BMI ≥ 23)* | *Obesity (BMI ≥ 27.5)* |
|---|---|---|---|---|---|
| **Male** | **Age 18–40 Years** | *Optimal cut-off point* | *≤26.5 cm* | *≥28.5 cm* | *≥29.5 cm* |
| | | AUC (%) (95% CI) | 82.1 (80.7-83.4) | 84.8 (83.5-86.0) | 89.4 (88.3-90.5) |
| | | Sensitivity (%) (95% CI) | 71.1 (67.2-74.8) | 77.6 (74.9-80.1) | 87.8 (82.3-92.0 |
| | | Specificity (%) (95%CI) | 77.8 (76.2-79.4) | 76.9 (75.0-78.6) | 76.6 (75.0-78.1)) |
| | | Youden Index | 0.489 | 0.545 | 0.643 |
| | **Age ≥ 60 Years** | *Optimal cut-off point* | *≤26.5 cm* | *≥28.5 cm* | *≥29.5 cm* |
| | | AUC (%) (95% CI) | 83.7 (81.6-85.6) | 83.7 (81.7-85.6) | 86.1 (84.2-87.9) |
| | | Sensitivity (%) (95% CI) | 70.5 (64.2-76.3) | 76.8 (73.0-80.3) | 81.4 (73.1-87.9) |
| | | Specificity (%) (95%CI) | 80.4 (78.0-82.7) | 75.4 (72.3-78.3) | 72.8 (70.3-75.3) |
| | | Youden Index | 0.510 | 0.522 | 0.542 |
| | **Age ≥ 60 Years** | *Optimal cut-off point* | *≤25.5 cm* | *≥27.5 cm* | *≥29.5 cm* |
| | | AUC (%) (95% CI) | 83.4 (81.0-85.6) | 84.3 (82.0-86.5) | 91.5 (89.7-93.2) |
| | | Sensitivity (%) (95% CI) | 66.8 (60.8-72.4) | 85.3 (80.7-89.1) | 85.4 (72.2-93.9) |
| | | Specificity (%) (95%CI) | 84.5 (81.8-87.0) | 70.3 (67.0-73.6) | 83.8 (81.4-86.0) |
| | | Youden Index | 0.513 | 0.556 | 0.692 |
| **Female** | **Age 18–40 Years** | *Optimal cut-off point* | *≤25.5 cm* | *≥27.5 cm* | *≥28.5 cm* |
| | | AUC (%) (95% CI) | 87.5 (86.6-8.5) | 86.2 (85.2-87.2) | 88.0 (87.1-89.0) |
| | | Sensitivity (%) (95% CI) | 82.4 (79.1-85.3) | 72.1 (70.2-74.0) | 82.3 (79.2-85.2) |
| | | Specificity (%) (95%CI) | 77.9 (76.6-79.2) | 82.0 (80.4-83.5) | 77.4 (76.0-78.7) |
| | | Youden Index | 0.603 | 0.541 | 0.597 |
| | **Age 40–60 Years** | *Optimal cut-off point* | *≤25.5 cm* | *≥26.5 cm* | *≥28.5 cm* |
| | | AUC (%) (95% CI) | 89.0 (87.5-90.4) | 86.4 (84.8-88.0) | 88.0 (86.4-89.5) |
| | | Sensitivity (%) (95% CI) | 85.2 (80.3-89.3) | 81.9 (79.3-84.3) | 79.1 (74.3-83.4) |
| | | Specificity (%) (95%CI) | 77.4 (75.2-79.4) | 73.1 (70.0-75.9) | 79.5 (77.3-81.5) |
| | | Youden Index | 0.626 | 0.550 | 0.586 |
| | **Age ≥ 60 Years** | *Optimal cut-off point* | *≤24.5 cm* | *≥26.5 cm* | *≥27.5 cm* |
| | | AUC (%) (95% CI) | 84.7 (82.1-87.0) | 86.5 (84.1-88.7) | 91.9 (89.9-93.6) |
| | | Sensitivity (%) (95% CI) | 75.1 (69.1-80.4) | 74.3 (68.7-79.3) | 87.1 (78.0-93.4) |
| | | Specificity (%) (95%CI) | 76.2 (72.7-79.5) | 84.3 (81.1-87.1) | 81.4 (78.5-84.0) |
| | | Youden Index | 0.513 | 0.586 | 0.684 |

BMI: body mass index; MUAC: mid-upper arm circumference; AUC: area under the curve.

## Discussion

The objective of the present study was to establish MUAC cut-off points to screen adult individuals with underweight, overweight, and obesity in Bangladesh, as proposed by the WHO in 2004 and Tang *et al*. in 2020 [9,21], as well as to establish sex- and age-specific cut-off points. The present study proposed age-specific MUAC cut-offs for males and females separately. The overall MUAC cut-off points for underweight, overweight and obesity in adult individuals varied across sex and age categories (Table 2). Considering this variation, the present study suggests different cut-off points for males and females in different age groups, which are suitable for field studies to correctly classify underweight, overweight, and obesity. The high sensitivity and specificity (Table 3) indicated the robustness and suitability of our findings for establishing MUAC as a surrogate measure of BMI in Bangladesh.

 

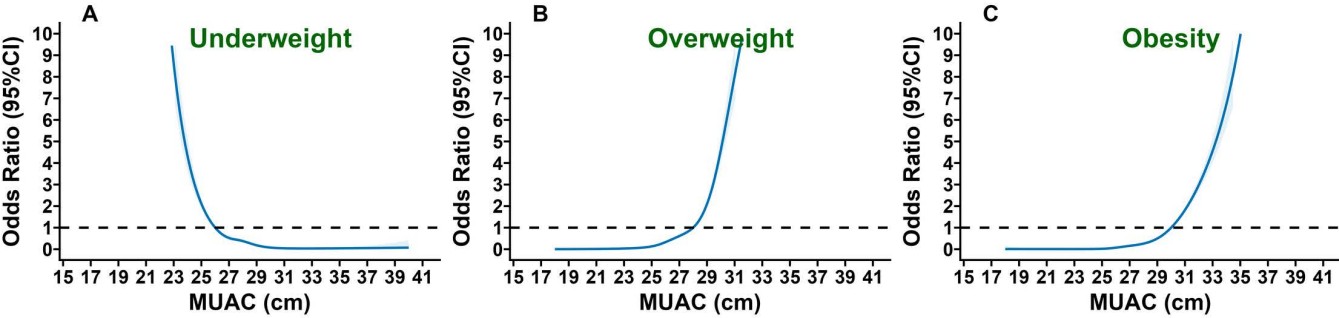

**Fig 2. Odds of being underweight (A), overweight (B) or obese (C) according to restricted cubic splines (RCSs) regression.**

Several previous studies have attempted to report MUAC cut-off points as substitutes for BMI [9,12,13]. However, most studies have focused on predicting underweight rather than overweight or obese. A recent large meta-analysis was conducted to establish global MUAC cut-off points for underweight individuals, and MUAC cut-off points ranging from ≤23·5 to ≤25·0 cm were used to indicate underweight [9]. A recent observational study from Bangladesh reported that a MUAC cut-off <25.1 cm in males and <23.9 cm in females is suitable for screening underweight [18], which is lower than the findings from the present study (males: ≤26.5 cm for younger and middle age and ≤25.5 cm for older; females: ≤25.5 cm for younger and middle age and ≤24.5 cm for older). The lower MUAC cut-off points in underweight in the previous study may be attributed to the significant differences in the study design and target population. In a previous study, data were collected from a single site, the sample size was small, and the data were collected from patients attending a tertiary hospital, rendering the findings unrepresentative of the adult population of the entire country. Furthermore, the population of interest was younger, with a lower average age compared to our study, and the proposed cut-off was sex-specific, however, unlike the current study, it was not stratified by age categories [18]. Additionally, a few studies from neighboring countries in India also reported a lower MUAC cut-off of 24 cm for undernutrition among adults [29,30], with 23 cm/24 cm in males and 22 cm in females [13,31]. However, the major limitations of these studies were the: small sample size, conducted on tribal population in slum areas where chronic energy deficiency was more common, and therefore lacking representativeness of the entire country. For individuals with chronic energy deficiency, BMI fails to account for low muscle mass or arm muscle area [32], which may be attributed to the lower MUAC cut-off for underweight in the neighboring country.

The present study demonstrated comparatively lower MUAC cut-off points for overweight and obesity (Table 3) than the findings from the neighboring country of India. The study from India reported a cut-off MUAC of 31.3 cm for males, 28.5 cm for females for screening overweight and 31.2 cm for males and 28.3 cm for females for screening for abdominal obesity [15]. Although the sensitivity and specificity in this study were similar, the cut-off points were greater than those found in our study. This variation in cut-off points is attributable to the use of WHO BMI cut-off points for overweight (BMI ≥ 25 kg/m²) in the Indian study, whereas the WHO BMI cut-off points for Asians were used in the present study (BMI ≥ 23.0 kg/m²). Furthermore, abdominal obesity rather than obesity based on BMI was considered in an Indian study. A limited number of studies from other parts of the world have proposed overall, sex-specific and age-specific cut-off values for overweight and obesity among adults, which are consistent with the findings of the current study [14,33–35].

The findings from the present study have implications for utilizing MUAC as a surrogate screening tool for underweight, overweight, and obese individuals in Bangladesh. In LMICs, such as Bangladesh, where economic and professional resources are limited, MUAC could be used as an inexpensive tool for determining underweight, overweight, and obesity. Traditional anthropometric indices, such as BMI, fail to assess body fat distribution or discriminate between visceral and subcutaneous adipose tissues, which can be explained by the term "Obesity Paradox". The obesity paradox

is a phenomenon in which a population with a normal BMI exhibits a higher risk of developing metabolic diseases and increased mortality, whereas people with obesity exhibit a lower risk. The distribution of fat rather than its mass is reported to explain the obesity paradox [36]. Although universally acceptable MUAC cut-off points for underweight, overweight, and obesity are yet to be established, these cut-off points are increasingly used because of their simplicity and portability.

The decreasing MUAC cut-off for underweight, overweight and obesity at a higher order of age in the current study is attributable to the negative association between BMI and MUAC at a higher order of age (S1 Table). These findings underscore the necessity for age-specific MUAC cutoff values. In older adults, lower MUAC cut-offs may be required because of the redistribution of adipose tissue from the extremities to the central body region with age. This physiological change can result in reduced MUAC, which may not accurately reflect an individual's nutritional status [23]. The upper arms contain muscle and fat mass, and their circumferences are reflected well in their changes [37]. The establishment of standardized MUAC cut-off values for adults who are underweight, overweight, and obese could enable the identification of beneficiaries at elevated risk of undernutrition or overnutrition. The prevalence of underweight, overweight and obesity measured by the proposed MUAC cut-off in the current study was comparable to that measured by the BMI category (S3 Table). These standardized the suitability of the proposed MUAC cut-off points to properly utilize limited resources to enhance public health management in developing countries such as Bangladesh.

The mid-upper arm circumference (MUAC) cut-offs identified in this study have potential as straightforward and cost-effective screening tools for detecting underweight and overweight/obese individuals in both clinical and community-based environments. In primary care or outreach initiatives, MUAC can be swiftly measured without the need for scales or stadiometers, rendering it particularly advantageous in resource-constrained or emergencies. Nevertheless, prior to the widespread adoption of these cut-offs, further research is imperative to validate their predictive accuracy across diverse populations, age groups, and clinical conditions. Longitudinal studies could also elucidate whether these thresholds consistently predict health outcomes such as metabolic syndrome, morbidity, and mortality.

A significant strength of this study is that, to the best of our knowledge, this is the first study to investigate MUAC cut-off points for screening overweight and obesity. Additionally, this study was conducted using a large representative national dataset encompassing the entire country, allowing for the generalization of the findings to the whole country. A limitation of this study is the reliance on a common index, namely, the BMI, to define underweight, overweight, and obesity. This may not adequately differentiate between appropriate fat and muscle mass. Some odds ratios reported in this study were notably large (S2 Table), particularly in the stratified analyses. These inflated estimates may have resulted from sparse data or low event rates in certain subgroups, leading to overestimation and wide confidence intervals. Therefore, further studies with larger sample sizes are warranted in each subgroup. Additionally, a limitation of the MUAC is its susceptibility to physiological and pathological conditions, such as peripheral edema, which may artificially increase arm circumference, whereas localized muscle hypertrophy could lead to an overestimation of nutritional reserves. Furthermore, the data utilized were secondary in nature and lacked information on inter- or intra-observer variability, which constrained the assessment of measurement variability. Moreover, the ethnic background of the study population was not considered in the current study, which is necessary to account for ethnic variations. Further empirical studies using gold-standard body composition analysis, such as dual-energy X-ray absorptiometry (DEXA), BodPod, or bioelectrical impedance analysis, covering people from different ethnic backgrounds are recommended to overcome these limitations.

## Conclusion

This study represents the first prospective investigation utilizing a large representative database to explore the validity of MUAC as a single anthropometric measurement for screening underweight, overweight and obese individuals in Bangladesh. The current study recommends sex- and age-specific MUAC cut-off points as alternatives to BMI for community-based screening of underweight, overweight, and obese individuals in Bangladesh (Table 3 and S1 Fig). As a simple and inexpensive procedure, MUAC is well suited for extensive population surveys in resource-limited countries,

such as Bangladesh. To enhance its utility, MUAC can be integrated into existing health systems through the implementation of routine screening protocols at primary care facilities and outreach clinics. Furthermore, incorporating MUAC into national nutrition surveillance programs would enable efficient, large-scale monitoring of undernutrition and overnutrition.

## Supporting information

**S1 Fig. Graphical abstract.**
(TIF)

**S1 Table. Multiple linear regression outcome of association between MUAC and BM.**
(PDF)

**S2 Table. Likelihood of underweight, overweight and obesity by MUAC quartile.**
(PDF)

**S3 Table. Morbidity (diabetes and hypertension) prevalence by proposed MUAC cut-off points by gender and age category.**
(PDF)

## Acknowledgments

The researchers express their gratitude to the Demographic and Health Survey (DHS) authority and ICF international for making the data accessible to the scientific community for analysis and publication. The study participants are also acknowledged with appreciation.

## Author contributions

**Conceptualization:** Sheikh Arafat Rahman, Md. Humayan Kabir, Shaikh Shahinur Rahman, Md Kamruzzaman.

**Data curation:** Sheikh Arafat Rahman, Md. Humayan Kabir, Md Kamruzzaman.

**Formal analysis:** Sheikh Arafat Rahman, Md. Humayan Kabir, Md Kamruzzaman.

**Methodology:** Md Kamruzzaman.

**Software:** Md Kamruzzaman.

**Supervision:** Md Kamruzzaman.

**Writing – original draft:** Sheikh Arafat Rahman, Md. Humayan Kabir, Shaikh Shahinur Rahman, Md Kamruzzaman.

**Writing – review & editing:** Sheikh Arafat Rahman, Md. Humayan Kabir, Shaikh Shahinur Rahman, Md Kamruzzaman.

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
