## [Decision Letter · Decision Letter 0]

PONE-D-25-18917The validity of mid-upper arm circumference as an indicator of underweight, overweight and obesity adults in BangladeshPLOS ONE

Dear Dr. Kamruzzaman,

Thank you for submitting your manuscript to PLOS ONE. After careful consideration, we feel that it has merit but does not fully meet PLOS ONE’s publication criteria as it currently stands. Therefore, we invite you to submit a revised version of the manuscript that addresses the points raised during the review process.

**ACADEMIC EDITOR: Experts in the field have reviewed your manuscript and you are expected to address their comments as early as possible. Thank you./>==============================**

**Please submit your revised manuscript by ** Jul 18 2025 11:59PM**. If you will need more time than this to complete your revisions, please reply to this message or contact the journal office at plosone@plos.org . **

**A rebuttal letter that responds to each point raised by the academic editor and reviewer(s). You should upload this letter as a separate file labeled 'Response to Reviewers'.****A marked-up copy of your manuscript that highlights changes made to the original version. You should upload this as a separate file labeled 'Revised Manuscript with Track Changes'.****An unmarked version of your revised paper without tracked changes. You should upload this as a separate file labeled 'Manuscript'.**

****

We look forward to receiving your revised manuscript.

**Kind regards,**

**Olutosin Ademola Otekunrin**

**Academic Editor**

PLOS ONE

2. In the online submission form, you indicated that your data will be submitted to a repository upon acceptance.  We strongly recommend all authors deposit their data before acceptance, as the process can be lengthy and hold up publication timelines. Please note that, though access restrictions are acceptable now, your entire minimal  dataset will need to be made freely accessible if your manuscript is accepted for publication. This policy applies to all data except where public deposition would breach compliance with the protocol approved by your research ethics board. If you are unable to adhere to our open data policy, please kindly revise your statement to explain your reasoning and we will seek the editor's input on an exemption.

****

Reviewers' comments:

**Reviewer's Responses to Questions**

**Comments to the Author**

1. Is the manuscript technically sound, and do the data support the conclusions?

**Reviewer #1: Yes**

**Reviewer #2: Yes**

**2. Has the statistical analysis been performed appropriately and rigorously? **

**Reviewer #1: I Don't Know**

**Reviewer #2: Yes**

3. Have the authors made all data underlying the findings in their manuscript fully available?

**Reviewer #1: Yes**

**Reviewer #2: Yes**

**4. Is the manuscript presented in an intelligible fashion and written in standard English?**

**Reviewer #1: Yes**

**Reviewer #2: Yes**

**5. Review Comments to the Author**

**Reviewer #1: This manuscript presents a well-conducted analysis using a nationally representative dataset (BDHS 2017–18) to propose sex- and age-specific mid-upper arm circumference (MUAC) cut-offs for identifying underweight, overweight, and obesity among adults in Bangladesh. The methods are statistically sound, and the manuscript is generally well-organized. However, several areas could be improved to strengthen the overall quality and clarity of the paper. As I do not have sufficient expertise in the advanced statistical methods applied in this study, I recommend that a statistician review the analytical components to ensure their appropriateness and robustness.**

Introduction:

1.The introduction is well-written, but could be made more concise by removing some repetitive information.

2.While the strengths of MUAC are discussed, its limitations are not adequately addressed. For example, MUAC measurements may be less accurate in individuals with conditions like edema or muscle hypertrophy or variations in measurement technique. In some circumstances, it might be best to complement MUAC with other anthropometric indicators to provide a more complete picture.

Methods:

1.Please justify the selection of age groups (18–40, 40–60, >60). Were these based on previous studies, expert opinion, or data-driven criteria?

2.Line 222: The Youden Index is mentioned as the method for determining cut-off points. Please ensure consistent spelling throughout the text.

3.Lines 227–230: Restricted cubic spline analysis is appropriate, but the number and placement of knots should be explained, e.g., was it based on the AIC criteria?.

4.Line 237: Please use decimal points (.) consistently in accordance with English language conventions, rather than commas (,).

5.Please indicate whether any data were missing and how missing values were handled.

6.Consider moving the ethical approval and data access information into a separate subsection titled “Ethics Approval and Data Availability” to improve clarity and organization.

Results:

1.The Results section is quite dense and could be difficult to follow. Adding subheadings for different analysis components could enhance readability.

2.Please define all abbreviations in tables and figures, including “+PV” and “–PV,” in the footnotes to ensure they are self-explanatory.

Discussion:

1.The discussion does not explain why MUAC cut-offs are lower in older adults or the biological basis of sex-specific differences. Please consider discussing age-related muscle mass loss and fat redistribution as possible explanations.

2.Some odds ratios reported (e.g., OR > 100) appear extremely large. It would be helpful to caution readers about potential overestimation due to stratification or sparse data.

3.Consider elaborating on how these MUAC cut-offs could be used in clinical and community settings and whether further research is needed to validate their use.

4.The discussion could be expanded to include examples of how MUAC could be applied in the field, such as for community health worker (CHW) screening or integration into national nutrition surveillance systems.

5.Several relevant limitations are not mentioned, including: (a) MUAC measurements can be distorted by conditions such as edema or muscle hypertrophy, or (2) No discussion is provided on measurement reliability (e.g., inter- or intra-observer variability), which is important for field-based anthropometric tools like MUAC.

Conclusions:

1.The conclusion notes the use of MUAC but could be expanded by adding some suggestions for how it could be integrated into health systems or programs.

2.Authors could add a sentence about whether there is a need for validation with body composition measures and implementation studies.

**Reviewer #2: I suggest to add to abstract that the cut-off point were selected by Younden index.**

**Statistic**

I suggest to summarize it to avoid repetition.

I suggest to add the word adults through out the work

I failed to see if you include or exclude pregnant women

**6. PLOS authors have the option to publish the peer review history of their article (what does this mean? ). If published, this will include your full peer review and any attached files.**

**Do you want your identity to be public for this peer review?** For information about this choice, including consent withdrawal, please see our Privacy Policy . 

**Reviewer #1: No**

**Reviewer #2: No**

****

---

## [Author Response · Author response to Decision Letter 1]

6 Jun 2025

Response to the reviewer’s comments

Reviewer 1:

Introduction:

1. The introduction is well-written but could be made more concise by removing some repetitive information.

Response: Repetitive information has been removed from introduction and corrected.

(Page 4)

2. While the strengths of MUAC are discussed, its limitations are not adequately addressed. For example, MUAC measurements may be less accurate in individuals with conditions like edema or muscle hypertrophy or variations in measurement technique. In some circumstances, it might be best to complement MUAC with other anthropometric indicators to provide a more complete picture.

Response: Thanks for this comment. Limitations of MUAC has been added in the introduction section.

(Page 5, Line 120)

Methods:

1. Please justify the selection of age groups (18–40, 40–60, >60). Were these based on previous studies, expert opinion, or data-driven criteria?

Response: We thank the reviewers for this observation. The chosen age groups (18–40, 40–60, >60 years) were determined based on a combination of physiological relevance and alignment with existing literature concerning age-related variations in body composition and nutritional risk.

(Page 8, Line 215)

2. Line 222: The Youden Index is mentioned as the method for determining cut-off points. Please ensure consistent spelling throughout the text.

Response: Thanks for this comment. Youden index has been corrected consistently.

(Page 9 Line 240)

3. Lines 227–230: Restricted cubic spline analysis is appropriate, but the number and placement of knots should be explained, e.g., was it based on the AIC criteria?

Response: Thanks for this important comment. The number of knots and model fit has now added the methodology section.

(Page 10, Line 244)

4. Line 237: Please use decimal points (.) consistently in accordance with English language conventions, rather than commas (,).

Response: Thanks for this comment. Decimal point has been corrected now.

(Page 10, Line 256)

5. Please indicate whether any data were missing and how missing values were handled.

Response: Missing data are included in the methodology section.

(Page 7, Line 179)

6. Consider moving the ethical approval and data access information into a separate subsection titled “Ethics Approval and Data Availability” to improve clarity and organization.

Response: Thanks for this comment. Ethics Approval and Data Availability section has been separated.

(Page 7, Line 185)

Results:

1. The Results section is quite dense and could be difficult to follow. Adding subheadings for different analysis components could enhance readability.

Response: Thanks for this comment. We have now divided the results into different subsections.

2. Please define all abbreviations in tables and figures, including “+PV” and “–PV,” in the footnotes to ensure they are self-explanatory.

Response: All abbreviations including +PV and -PV has been abbreviated.

(Page 13, Line 309)

Discussion:

1. The discussion does not explain why MUAC cut-offs are lower in older adults or the biological basis of sex-specific differences. Please consider discussing age-related muscle mass loss and fat redistribution as possible explanations.

Response: Thanks for this comment. Causes of lower MUAC cut-offs points in older people has now been discussed in the discussion section.

(Page 18, Line 422)

2. Some odds ratios reported (e.g., OR > 100) appear extremely large. It would be helpful to caution readers about potential overestimation due to stratification or sparse data.

Response: We appreciate the reviewer's helpful comment regarding the very large odds ratios (OR >100) found in some subgroup analyses. We agree that these might show sparse data or a few events, which can lead to unstable or exaggerated estimates. This has now been acknowledged as a limitation of data sparse bias.

(Page 19 Line 454)

3. Consider elaborating on how these MUAC cut-offs could be used in clinical and community settings and whether further research is needed to validate their use.

Response: We appreciate this valuable suggestion. We have expanded the practical implications of our MUAC cut-offs, highlighting their potential usefulness in both clinical and community health environments. In addition, we recognize the necessity for further validation studies to verify the generalizability and prognostic value of these thresholds. This information has been included in the discussion section.

(Page 18, Line 437)

4. The discussion could be expanded to include examples of how MUAC could be applied in the field, such as for community health worker (CHW) screening or integration into national nutrition surveillance systems.

Response: We appreciate the reviewer's helpful suggestion. We have added more details to the discussion section. Now, we include examples of how MUAC can be used in real-life situations.

(Page 18, Line 437)

5. Several relevant limitations are not mentioned, including:

o MUAC measurements can be distorted by conditions such as edema or muscle hypertrophy.

o No discussion is provided on measurement reliability (e.g., inter- or intra-observer variability), which is important for field-based anthropometric tools like MUAC.

o

Response:

o We appreciate this comment. Effect of edema or muscly hypertrophy on MUAC measurement has been acknowledged in the limitations section.

(Page 19, Line 459)

o Thanks for tis comment. The data, that we have used, are secondary in nature and in the data definition, there are no information on inter or intra observation variability. Therefore, we are unable discuss on measurement reliability. However, we have acknowledged that in the limitation section.

(Page 19, Line 462)

Conclusions:

1. The conclusion notes the use of MUAC but could be expanded by adding some suggestions for how it could be integrated into health systems or programs.

Response: We appreciate this comment. Use of MUAC has been expanded in the conclusion section.

(Page 20, Line 479)

2. Authors could add a sentence about whether there is a need for validation with body composition measures and implementation studies.

Response: We appreciate this comment. Need for validation with body composition measures and implementation studies has already been added in the discussion section.

(Page 19, Line 466)

Reviewer 2:

1. I suggest to add to abstract that the cut-off point were selected by Younden index.

Response: Cut-oof based on Youden index has been added now.

(Page 2, Line 52)

2. I suggest to summarize it to avoid repetition.

Response: Introduction has been edited to avoid repetition.

3. I suggest to add the word adults throughout the work.

Response: Word adults has been added throughout the work.

4. I failed to see if you include or exclude pregnant women.

Response: Thanks for this comment. Inclusion of non-pregnant has been added in the methodology section.

(Page 7, Line 180)

---

## [Decision Letter · Decision Letter 1]

The validity of mid-upper arm circumference as an indicator of underweight, overweight and obesity adults in Bangladesh

PONE-D-25-18917R1

Dear Dr. Kamruzzaman,

We’re pleased to inform you that your manuscript has been judged scientifically suitable for publication and will be formally accepted for publication once it meets all outstanding technical requirements.

Kind regards,

Olutosin Ademola Otekunrin

Academic Editor

PLOS ONE

Additional Editor Comments (optional):

Reviewers' comments:

Reviewer's Responses to Questions

**Comments to the Author**

1. If the authors have adequately addressed your comments raised in a previous round of review and you feel that this manuscript is now acceptable for publication, you may indicate that here to bypass the “Comments to the Author” section, enter your conflict of interest statement in the “Confidential to Editor” section, and submit your "Accept" recommendation.

Reviewer #2: All comments have been addressed

2. Is the manuscript technically sound, and do the data support the conclusions?

Reviewer #2: Yes

3. Has the statistical analysis been performed appropriately and rigorously? 

Reviewer #2: Yes

4. Have the authors made all data underlying the findings in their manuscript fully available?

Reviewer #2: Yes

5. Is the manuscript presented in an intelligible fashion and written in standard English?

Reviewer #2: Yes

6. Review Comments to the Author

Reviewer #2: Authors have responded and replied to all of my comments.

I suggest to accept this work in its current form.

7. PLOS authors have the option to publish the peer review history of their article (what does this mean? ). If published, this will include your full peer review and any attached files.

**Do you want your identity to be public for this peer review?** For information about this choice, including consent withdrawal, please see our Privacy Policy .

Reviewer #2: No

---

## [Editor Report · Acceptance letter]

PONE-D-25-18917R1

PLOS ONE

Dear Dr. Kamruzzaman,

I'm pleased to inform you that your manuscript has been deemed suitable for publication in PLOS ONE. Congratulations! Your manuscript is now being handed over to our production team.

Kind regards,

on behalf of

Dr. Olutosin Ademola Otekunrin

Academic Editor

PLOS ONE